# Characterization of Rhizosphere and Endophytic Microbial Communities Associated with *Stipa purpurea* and Their Correlation with Soil Environmental Factors

**DOI:** 10.3390/plants11030363

**Published:** 2022-01-28

**Authors:** Haoyue Liu, Jinan Cheng, Hui Jin, Zhongxiang Xu, Xiaoyan Yang, Deng Min, Xinxin Xu, Xiangfeng Shao, Dengxue Lu, Bo Qin

**Affiliations:** 1CAS Key Laboratory of Chemistry of Northwestern Plant Resources/Key Laboratory for Natural Medicine of Gansu Province, Lanzhou Institute of Chemical Physics, Chinese Academy of Sciences (CAS), Lanzhou 730000, China; yanzuhua21@mails.ucas.ac.cn (H.L.); yangxiaoy@licp.cas.cn (X.Y.); mindeng@licp.cas.cn (D.M.); XUXX@licp.cas.cn (X.X.); 2University of Chinese Academy of Sciences, Beijing 100049, China; 3State Key Laboratory of Grassland Agro-Ecosystems, College of Pastoral Agricultural Science and Technology, Lanzhou University, Lanzhou 730000, China; chengjn2019@lzu.edu.cn; 4Yantai Zhongke Research Institute of Advanced Materials and Green Chemical Engineering, Yantai 264006, China; 5Animal, Plant & Food Inspection Center of Nanjing Customs, Nanjing 210000, China; nj4183@nj.intra.customs.gov.cn; 6State Key Laboratory of Applied Organic Chemistry, Lanzhou University, Lanzhou 730000, China; liuhaoyue@licp.cas.cn; 7Institute of Biology, Gansu Academy of Sciences, Lanzhou 730000, China; gskxy@gsas.ac.cn

**Keywords:** *S. purpurea*, microbial community, soil environmental factors, high-throughput sequencing, Qinghai-Tibetan Plateau

## Abstract

This study was to explore the diversity of rhizosphere and endophytic microbial communities and the correlation with soil environmental factors of *Stipa purpurea* on the Qinghai-Tibetan Plateau. The bacterial phylum of Proteobacteria, Firmicutes and Bacteroidota, and the fungal phylum of Ascomycota, Basidiomycota and Zygomycota were dominant in microbial communities of *S. purpurea* in all three sampling sites. Multiple comparison analysis showed that there were significant differences in the composition of microbial communities in the roots, leaves and rhizosphere soil. Whether it is fungi or bacteria, the OTU abundance of rhizosphere soils was higher than that of leaves and roots at the same location, while the difference among locations was not obvious. Moreover, RDA analysis showed that Zygomycota, Cercozoa, Glomeromycota, Chytridiomycota and Rozellomycota possessed strongly positive associations with altitude, dehydrogenase, alkaline phosphatase, neutral phosphatase, available kalium and available phosphate, while Ascomycota was strongly negatively associated. Changes in ammonium nitrate, alkaline phosphatase, polyphenol oxidase, total phosphorus, and altitude had a significant impact on the bacterial communities in different habitats and altitudes. Taken together, we provide evidence that *S. purpurea* has abundant microbial communities in the alpine grassland of the Qinghai-Tibetan Plateau, whose composition and diversity are affected by various soil environmental factors.

## 1. Introduction

Microorganisms associated with plants, mostly endophytic and rhizosphere soil microorganisms, are important components of biodiversity in ecosystems and play crucial roles on plant health and soil fertility [1]. Most endophytic microorganisms can enhance the resistance of the host plant to diseases, drought and abiotic stresses, which are widely valued in agriculture and animal husbandry [2]. Studies have shown that the greatest impact of endophytic microorganisms on the physiological and ecological effects of plant tissues and organs is on root tissue, followed by leaf tissue [3]. Soil microorganisms serve as the largest resource pool of the grassland underground ecosystem. An increasing number of studies have focused on the rhizosphere microbial community, which is defined as the soil closely connected to the roots (within 2.5 mm) [4]. Importantly, rhizosphere microorganisms can directly or indirectly affect plant growth and produce beneficial, neutral or harmful substances [5]. For example, rhizosphere microorganisms can enhance the resistance of plants to biotic and abiotic stresses by producing growth-promoting substances and inhibiting pathogens in the soil [6]. Rhizosphere microorganisms are sensitive to changes in soil quality, especially soil microbial diversity as a sensitive indicator of soil biological properties, which can indicate early changes in grassland ecological environment and changes in ecosystem functions [7]. Therefore, it is important to study the microbial community structure and diversity for grassland ecosystem protection, restoration and reconstruction. 

Soil physical and chemical properties and enzyme activities are important factors that affect the composition and diversity of microbial communities. In one study, soil microbial diversity [8] and community structure [9] were distinctly different in different altitude gradients, which were significantly related to the C/N ratio or soil pH. Likewise, other factors that can rapidly affect microbial communities in the Qinghai-Tibet Plateau (QTP) include carbon, phosphorus and nitrogen contents [10,11], and their structure and abundance change due to their high sensitivity to environmental conditions [9].

*Stipa purpurea* is a perennial xerophytic grass that belongs to the Gramineae family, it has a wide range of distributions from the QTP, Pamir Plateau and central Asian alpine region, and is mainly distributed in Gansu, Qinghai, Xinjiang, Sichuan and Tibet of China [12]. Because *S. purpurea* has a strong adaptability to stress conditions including low temperature, drought, poisonous weed invasion and barren soil in the plateau environment, it has become a constructive species and dominant population of alpine grasslands. Moreover, the stems and leaves have good palatability, are rich in protein and fat, have high nutritional value, and are palatable to livestock. As one of the most important high-quality grass resources, *S. purpurea* provides an important material foundation for the development of animal husbandry. At present, studies on *S. purpurea* have mainly focused on genetic diversity [13], community classification [14], grazing effects [15], environmental response [16] and so on. In addition, research on rhizosphere and endophytic microorganisms in *S. purpurea* grasslands has also attracted much attention. Lu et al. [17] studied the fungal community structure in the rhizosphere and roots of *S. purpurea* in Qinghai by ITS library construction. Bao et al. [18] showed that there was no obvious specificity between the *S. purpurea* endophytic fungus and the host. Short-term warming had no significant effect on soil microbial biomass and community structure based on the phospholipid fatty acids (PLFA) method [19].

In this study, we evaluated the influence of soil physical and chemical properties and enzyme activity on the microbial community characteristics of *S. purpurea* from three sites located in the QTP, including Qinghai, Gansu, and Tibet. Firstly, the physical and chemical properties and enzyme activities of rhizosphere soil were determined using soil chemistry methods. Then, the composition and diversity of the microbial community in rhizosphere and endophytic microbial communities were analyzed by high throughput sequencing. Finally, the correlation analysis method was used to evaluate the correlation between the *S. purpurea* microbial community and soil environmental factors. The desired targets of this work were as follows: (i) endophytic and rhizosphere soil microbial communities have divergent structures due to different habitats; (ii) environmental factors driving microbial community structure and diversity of *S. purpurea*.

## 2. Materials and Methods

### 2.1. Study Sites and Sample Collection

Rhizosphere soil and plant samples were collected in June 2021 in three locations: Zhuaxixiulong Township, Tianzhu Tibetan Autonomous County, Wuwei City, Gansu Province (102°51′10″ E, 37°09′07″ N), Qilian County, Haibei Prefecture, Qinghai Province (100°52′1″ E; 37°58′29″ N), Mozhugongka County, Lhasa city, Tibet (91°40′13″ E; 29°46′30″ N). The division of the sample plot refers to the method of [20]. Five healthy plants and rhizosphere soil were randomly collected in each plot; the soil closely attached to the roots (within 2.5 mm) was collected and regarded as rhizosphere soil [21]. All samples were immediately transported to the laboratory, packed in aseptic bags, and stored at 4 °C until processing.

Rhizosphere soil samples were obtained by brushing soil that attached to *S. purpurea* roots with sterile brush as described by [12]. It was divided into two parts, one for analysis of soil physicochemical properties and enzyme activity, and the other was stored in a refrigerator at −80 °C for the rhizosphere soil DNA extraction.

The roots and leaves were isolated from healthy *S. purpurea* plants with sterile scissors and cut into approximately 3 cm segments. It was washed with flowing tap water, soaked in 75% ethanol for 1 min, transferred to 2% sodium hypochlorite solution for 8 min, and finally cleaned with sterile water 5 times. The root and leaf samples disinfected on the surface were dried with sterile filter paper for use.

### 2.2. Determination of Soil Physicochemical Properties

Soil pH was determined by the mixed suspension of dry soil and deionized water (soil:water = 1:5 (w:v)) using a pH meter [22]. Potassium dichromate titration was used to assay soil organic matter (OM). Soil available P (AP) and available K (AK) were determined using NaHCO_3_ and ammonium acetate [23]. The total phosphorus (TP) of the soil was measured using the acid dissolution-molybdenum antimony anti-spectrophotometric technique. Soil ammonium nitrogen (AN) and nitrate nitrogen (NN) were detected by the KCl extraction indophenol blue colorimetry and the phenol disulfonic acid method, respectively [24]. Total nitrogen (TN) was assayed via H_2_SO_4_-HClO_4_ digestion.

### 2.3. Determination of Soil Enzymatic Activities

The activity of total phosphatase was measured colorimetrically using disodium phenyl phosphate, which is based on the hydrolysis reaction of disodium phenyl phosphate by measuring the amount of phenol produced [25]. The catalase (CAT) activity of the soil was determined through a potassium permanganate titration [26]. Sucrase (SC) and cellulase (CL) activity were detected by colorimetric analysis of 3,5-Dinitrosalicylic acid (DNSA) [27]. The activity of urease (UE) was detected through the method colorimetry of sodium phenate-sodium hypochlorite [28]. Dehydrogenase (DHA) activity was assayed by the reduction method of triphenyl tetrazolium chloride [29]. The activity of polyphenol oxidase (PPO) and peroxidase (POD) were determined by pyrogallol colorimetry [12].

### 2.4. Genomic DNA Extraction and PCR Amplification

The rhizosphere soil samples were frozen at −80 °C immediately after collection. The samples of the root and leaf were surface sterilized and stored at 4 °C. Following the manufacturer’s instructions, total DNA was extracted with a MagPure Soil DNA LQ Kit (Magen, Guangzhou, China) and a DNeasy PowerSoil Kit (QIAGEN, Hilden, Germany), respectively. DNA integrity was detected by 0.8% agarose gel electrophoresis, and concentration was measured via NanoDrop (Thermo Fisher, Shanghai, China) 2000 spectrophotometer. PCR amplification of the fungal ITS gene and bacterial 16S rRNA gene were performed using the ITS1F (5′-CTTG GTCA TTTA GAGG AAGT AA-3′), and ITS2R (5′-GCTG CGTT CTTC ATCG ATGC-3′) [30] primer pairs and 343F (5′-TACG GRAG GCAG CAG -3′) and 798R (5′- AGGG TATC TAAT CCT-3′) [31] primer pairs, respectively. The quality of the amplified product was observed by gel electrophoresis, purified with AMPure XP beads (agcourt), and another round of PCR was performed after amplification. After purification with AMPure XP beads again, the final amplicons were quantified with a Qubit dsDNA kit. Equal amounts of purified amplicon were pooled for subsequent sequencing.

### 2.5. Illumina MiSeq Sequencing and Processing

Sequencing was completed by the Illumina MiSeq platform. Then, paired-end reads were pretreated to detect and cut off the ambiguous bases (N) using Trimmomatic software [32], which also removed low quality sequences with average quality scores less than 20 using the sliding window trimming method. Sequences were depleted of barcodes and primers, short sequences <200 bp were removed, ambiguous and homologous sequences were removed, and sequences with a single base repeat greater than 8 were removed. Meanwhile, the sequences were then denoised and chimeras were detected and removed using QIIME software [33] (version 1.8.0).

After the removal of primer sequences, operational taxonomic units (OTUs) were generated at the level of 97% similarity by Vsearch software [34] (version 2.4.2). The representative read of each OTU was selected using the QIIME package. All representative reads were annotated and blasted against Silva database Version 138 (16s/18s rDNA) using RDP classifier [35], and against Unite database (ITSs rDNA) using BLAST [36], respectively.

### 2.6. Bioinformatics and Statistical Analysis

Pearson correlation analysis was completed with SigmaPlot 12.5 to evaluate the relationship between rhizosphere and endophytic microbial abundance and soil environmental factors. To identify significant differences and the correlation in fungal and bacterial community composition at different habitats (rhizosphere soil, leaves and roots), A Kruskal–Wallis one-way analysis of variance (one-way ANOVA) was used for ranking, then all pairwise multiple comparison procedures (Tukey test), and Spearman rank-order correlation were used to assess the correlation. The nonmetric multidimensional scaling (NMDS) plots were based on the analysis of the microbial phyla level using UniFrac distance metrics in R software. The RDA analyses were carried out using CANOCO 5 for Windows (Biometris, Wageningen, the Netherlands) to explore the correlation among microorganisms, physicochemical properties and soil enzyme activities.

## 3. Results

### 3.1. Soil Physicochemical Properties Subsection

The soil physicochemical properties of the three groups of soil were contrasted and analyzed in Figure 1. The results showed that the pH of the soil at three sites was neutral, partially alkaline, but the lowest pH was found at the QH site compared to XZ and GS (*p* < 0.01) (Figure 1d). The contents of TN (*p* < 0.05) and NN (*p* < 0.01) in the QH plot were considerably higher than those in GS and XZ (Figure 1c,g). The AK content of group GS was extremely lower than that the other two groups (*p* < 0.01) (Figure 1f). The AP content of XZ was considerably higher than GS (*p* < 0.05) (Figure 1e). However, there was no significant difference in OM, AN and TP contents among all groups (Figure 1a,b,h).

### 3.2. Soil Enzymatic Activity

The soil enzyme activities are shown in Figure 2. No significant difference was discovered among groups about the CL activity (Figure 2d) and UE activity (Figure 2e). When compared with the GS and QH groups, significant differences were found in that the activities of PPO (Figure 2a), DHA (Figure 2j) and ALP (Figure 2f) were significantly increased (*p* < 0.01) in XZ, while CAT activity was considerably lower (Figure 2b) (*p* < 0.05). The ALP activity was found to be directly positively related to altitude (*p* < 0.01) (Figure 2f). The POD (Figure 2c) and SC (Figure 2i) activities of group GS were considerably higher than those the other two groups (*p* < 0.05) and were extremely lower than the NP (Figure 2h) (*p* < 0.01). The ACP activity of QH was considerably lower than GS and XZ (*p* < 0.05) (Figure 2g).

### 3.3. Sequencing and Microbial Community Alpha Diversity

After processing, 589,686 and 542,759 high quality sequences of fungi and bacteria were retained in all 27 samples, respectively. The number of valid tags per sample ranged from 53,525 to 66,549. After clustering with a similarity level of 97%, a total of 21039 OTUs was generated (Table 1). The total OTUs comprised 2768, 667 and 454 for fungi in the rhizosphere soil, root and leaf, respectively. Likewise, the bacterial OTUs comprised 9258, 2882 and 5010 in the rhizosphere soil, root and leaf, respectively. Whether it was fungi or bacteria, the OTU abundance of rhizosphere soils was higher than that of leaves and roots at the same location, while the differences of microbial communities associated with the same habitats from different locations were not obvious. The rarefaction curves of 27 samples gradually approached the saturation (Figure 3). This indicated that the amount of data sequenced in this experiment was reasonable and that it fully reflected the composition of microbial community. Interestingly, whether it was fungi or bacteria, the amount of OTUs in the rhizosphere soil at the same location was higher than that of leaves and roots, while the difference among locations as not obvious. The Alpha diversity analysis indicated that the diversity of fungi and bacteria was abundant in *S*. *pur**purea* (Table 1). The diversity and richness of the rhizosphere soil samples were higher than those of the root and leaf samples, with the exception of the Shannon index in the leaf of Qinghai samples in Bacteria. The Chao1, Simpson and Shannon index results in the fungi of the rhizosphere soil samples were significantly (*p* < 0.05) higher than that in the leaf and root, and leaf was the lowest.

### 3.4. The Degree of Microbial Species (OTU) Overlap and Dissimilarity in Microbial Community Composition within and between Habitats

In order to identify significant differences and the correlation in fungal and bacterial community composition at different habitats (rhizosphere soil, leaves and roots) and locations, the Kruskal–Wallis one-way analysis of variance (one-way ANOVA) was used for ranking, then all pairwise multiple comparison procedures (Tukey test), and Spearman rank-order correlation was used to assess the correlation. There was no significant difference in the composition of fungal communities in samples from same habitat at different locations (*p* > 0.05) (Table 2). For QH samples, a significant difference (*p* < 0.05) between rhizosphere soil and root samples was detected. For XZ samples, there was a significant difference (*p* < 0.05) between rhizosphere soil and root samples, and a significant difference *(p* < 0.05) between rhizosphere soil and leaf samples. In addition, Spearman’s rank-order correlation showed that the samples in different locations of the same habitat had extremely significant positive correlation (*p* < 0.01). Thus, different habitats may be the main factors affecting fungal community composition in *S*. *pur**purea*.

In bacterial leaf samples, GS was significantly different from QH and XZ, respectively (*p* < 0.05) (Table 3). Similarly, several significant differences, such as rhizosphere soil and leaf in QH samples (*p* < 0.05), leaf and root in XZ samples (*p* < 0.05) were detected. Through Spearman’s rank-order correlation, we found that there was a significant positive correlation between rhizosphere soil and leaf samples at different locations (*p* < 0.05). In the GS samples, the composition of bacterial community leaves were significant negatively correlated and positively correlated with rhizosphere soil and roots, respectively (*p* < 0.05). Likewise, the roots were significant negatively and positively correlated with those of rhizosphere soil and leaf sample at XZ, respectively (*p* < 0.01).

### 3.5. Microbial Community Similarities in Different Samples

The NMDS plots based on the abundances of OTUs clearly showed that the community composition of bacteria and fungi have significant differences among all samples (Figure 4). The composition of fungal communities of the root, leaf and rhizosphere soil samples were more distinctly clustered and separated from each other (Figure 4a). Based on NMDS analysis, we found that fungal community composition differed significantly between rhizosphere soil, leaves and the roots of *S. purpurea*, though the sample pairs were separated from the different locations. For bacteria, the leaf samples were scattered and highly differentiated, indicating that the bacterial community was significantly different among different locations (Figure 4b). However, the samples of rhizosphere soil were more highly separated from the other samples.

### 3.6. Microbial Community Composition Analysis

At the phylum level, samples from different locations and habitats were analyzed for microbial community composition (Figure 5a,b). For the fungal communities, Ascomycota was dominant in the rhizosphere soil, root and leaf, especially the leaves of Gansu and Qinghai, which represented 97.26% and 98.86% of the total fungal OTUs (Figure 5a). However, in root of Tibet, this phylum only accounted for 37.41%. In addition, we found that the fungal community composition of the rhizosphere soils were very similar, even if they came from different locations. According to Figure 5b, in all samples, 42 bacterial phyla were detected. The most abundant bacterial phyla were Proteobacteria (10.81–66.96%), Firmicutes (0.9–63.22%), and Bacteroidota (3.54–41.37%), which accounted for over 80% of all sample sequences. Firmicutes, as the second-most abundant bacterial community phyla, was present at its highest level (63.22%) in Gansu leaves and was the lowest in the rhizosphere soils (representing 0.90% in Gansu, 1.02% in Qinghai, 1.31% in Tibet) Compared with other samples, Qinghai leaves had the highest abundance of bacterial community composition at the phylum level. The total bacterial composition of the rhizosphere soil samples from different regions were similar, while the abundance of each phylum varied in each sample.

The top 30 fungi and bacteria genera with the highest abundance were selected (Figure 5c,d). At the fungal genus level, *Psathyrella*, *Mortierella*, *Epichlo*, *Cortinarius*, *Neoascochyta* and *Ophiocordyceps* were the most dominant genera in the samples (Figure 5c). The proportion of the two genera (*Mortierella* and *Cortinarius*) were gradually increased with the increase of growth altitude in rhizosphere soil sample. *Psathyrella* was relatively enriched only in root of the Qinghai (71.32%) and Tibet (77.12%) samples. *Epichlo* were detected in all the samples, but it accounted for 90.84% in leaf of Tibet. The most abundant bacterial genera detected included *Enterococcus*, *Bacteroides* and *Sphingomonas* (Figure 5d), representing 10.89%, 10.99% and 17.66% of the total bacterial OTUs, respectively. The bacterial compositions of yhr leaf and root samples were more different than that of rhizosphere soil at the genus level, especially group of Qinghai leaf. *Sphingomonas* was the main genus detected rhizosphere soil samples. *Enterococcus* were the main genera in the group of root in Tibet and leaf in Gansu (52.85%), representing 39.73% and 52.85%, respectively.

### 3.7. Relationship between Microbial Community and Environmental Variables

The three most abundant phyla and Shannon index results in the fungi and bacteria of the rhizosphere soil samples were selected for Pearson testing. OM, AK, CL, AN, TN, ALP, NN, DHA and AP were significantly correlated with most of the fungal phyla (Appendix A). Of these factors, Ascomycota had negative correlations with OM and AN in the Gansu (*p* < 0.05) and Tibet (*p* < 0.01) samples respectively, while it had a distinctly positive correlation with AK and CL in Qinghai (*p* < 0.05). The phylum of Basidiomycota in Gansu and Qinghai were significantly negatively correlated with AN, TN and ALP (*p* < 0.05). Zygomycota was significant negatively correlated with AK in Gansu (*p* < 0.01) and positively correlated with NN, but negatively correlated with DHA in Qinghai (*p* < 0.05), and strongly negatively correlated with AP in Tibet (*p* < 0.05). The Shannon diversity index was significantly negatively correlated with attitude in Qinghai, negatively correlated with ACP in Gansu, and positively correlated with AP in Tibet, indicating that these factors of soil had an important influence on the distribution and diversity of rhizosphere fungal communities. At the level of bacterial phylum, Proteobacteria, Firmicutes and Bacteroidota with an average relative abundance greater than 20% were significantly correlated with most of soil environmental factors (Appendix B). AP showed distinctly positive correlations with Proteobacteria and Bacteroidota in the Gansu samples (*p* < 0.05). ALP and CL were positively associated with Proteobacteria (*p* < 0.05) in Gansu and Qinghai, respectively. Firmicutes showed significantly positive correlations with PH (*p* < 0.05) and a distinctly negative correlation with OM (*p* < 0.05). SC was positively correlated with Bacteroidota in Qinghai (*p* < 0.05), whereas DHA was strongly negatively associated with Bacteroidota in Tibet (*p* < 0.01).

In order to understand the main influencing factors of microbial community structure of *S. purpurea*, we used the redundancy analysis method to test the relationship between the community structure, soil physicochemical properties and enzyme activity (Figure 6). The phylum levels showed significant differences between root, leaf and rhizosphere soil of *S. purpurea*. In contrast, the samples of rhizosphere soil were most affected by environmental factors, whether it was fungi or bacteria. At the fungal phylum level (Figure 6a), the phyla Zygomycota, Cercozoa, Glomeromycota, Chytridiomycota and Rozellomycota possessed a strongly positive associations with altitude, DHA, ALP, NP, AK and AN, while Ascomycota was strongly negatively associated. Additionally, Ascomycota was strongly positively associated with SC and ACP, whereas Basidiomycota and Zygomycota were negatively correlated with SC and ACP. AN was positively and negatively correlated in Basidiomycota and Ascomycota, respectively. Similar to fungal abundance, the relative bacterial abundance was affected by soil physicochemical properties and enzyme activity (Figure 6b). Changes in AN, ALP, PPO, DHA, AP, TP, SC, CAT and altitude had more obvious effects on the structure and composition of bacteria in different habitats and altitudes. However, the changes in POD, NN and pH had relatively little influence. Gemmatimonadota, Myxococcota, Actinobacteriota and Acidobacteriota showed a strongly positive association with altitude, DHA, ALP, NP, PPO, OM and AP, while Firmicutes was strongly negatively associated with them. Proteobacteria was distinctly positively correlated with altitude, AN, DHA, ALP and PPO, while there was positive correlation with CAT and SC.

## 4. Discussion

This study found that there were significant differences in microbial community structure in different tissues and locations of *S. purpurea*. To our knowledge, this is the first study to integrally examine the microbial communities from the leaves, roots and rhizosphere soils of *S. purpurea* contemporaneously in the QTP.

At the phylum level, Ascomycota and Basidiomycota were the dominant phyla associated with *S. purpurea*, being phyla that are typical of leaf, root and rhizosphere soil fungi. Similar results were found with previous studies on the *S. purpurea* of the QTP [12]. With the increasing degree of grassland degradation, Ascomycota and Basidiomycota were the dominant groups of soil fungi in different degraded grasslands, and both phyla participate in the carbon cycle by degrading organic matter [20]. In general, members of Ascomycota are mostly found in harsh habitats, while Basidiomycota members prefer environments with rich resources and high plant abundance [37]. These attributes explain that the relative abundance of Basidiomycota was significantly higher in the root and rhizosphere soil than in leaf.

The dominant phyla of bacterial communities were basically consistent across the same habitats, but the relative abundances were obviously different, and varied with the types of dominant groups. Proteobacteria and Bacteroidota were widely distributed in all samples, while the phylum of Firmicutes were mainly distributed in the leaves and roots of *S. purpurea*. Firmicutes are widely distributed in a variety of habitats, which may be on account of their capacity to convert to a dormant form under stressful environment [38]. However, Firmicutes only occupied 0.43%, 0.49% and 0.63% of the rhizosphere soil samples in Gansu, Qinghai and Tibet, respectively. Proteobacteria, including a wide variety of pathogens, have also been found to be abundant in other QTP plants, such as *Stellera chamaejasme L*. [39,40]. We also found that under the conditions of human disturbance and climate change, the dominant components of soil bacteria and fungi communities under different altitudes of *S. purpurea* in the QTP were relatively stable.

Understanding the environmental variables that control the diversity and structure of microbial communities is one of the core objectives of microbial ecology [41]. The diversity of bacterial communities were observed to be more than that those of fungi, regardless of the microorganisms of rhizosphere or endophytes in all three locations in this study (Table 1). This confirms that moist habitats contribute to the growth of bacteria, while dry habitats are is conducive to the growth of fungi [42]. This may be related to the perennial snow on the QTP. In addition, we also found that the species richness and diversity of the rhizosphere soil community in the root was higher than in the leaves and roots, whether it was fungi or bacteria. Our findings, however, did not agree with the result obtained from [17] (Lu et al., 2016) research indicating that the fungal community diversity of *S. purpurea* roots is higher than that of rhizosphere soil in alpine steppes. This may have been due to the use of high-throughput sequencing methods and the samples from different sites on the QTP, which can explore the panorama of soil microbial diversity. In our study, 589,686 valid sequences were obtained, while only ~250 were found in the study of [17].

Generally, through research on the endophytic microbes in the leaves and roots and rhizosphere microbial communities of the dominant pasture *S. purpurea* at different locations in the QTP, it was found that habitat is the most important factor affecting the microbial diversity of *S. purpurea*. The difference in spatial niche is the main factor that determines the prokaryotic microbial community in leaves, roots and soil [43]. In addition, there were significant differences in the Shannon index among different habitats of the same location (*p* < 0.01) (Table 1). Our conclusion is in agreement with the study on different parts of soil, leaf, root and stem of poplar [44], which found that there are significant differences in the microbial community structure in the leaves and roots of the three *Agave* of the same genus. Among them, the contribution of plant habitats to the difference in microbial community structure is as high as 47.9%, while the contribution of plant types is 10.8% [45]. Plants can control and select microorganisms through a series of means, such as rhizosphere secretion [5], nutrient availability [46] and environmental differences [47]. Secondly, they will also be greatly affected by climate, physical and chemical properties, and other factors [48], which lead to the differences in the selection of microorganisms in leaves, roots and soil.

In our study, RDA correlation analysis showed that the relationship between the structure of bacterial and fungal communities at the phylum level and the measured physical and chemical parameters and enzyme activity parameters (Figure 6). It can be seen that, in the natural ecosystem of the QTP, the factors regulating the abundance of the *S. purpurea* microbial community were quite complex, and may be the result of a combination of multiple factors. Rhizosphere microorganisms are most affected by environmental factors, whether they are fungi or bacteria. This is consistent with the findings of previous studies [49]. Altitude, DHA, PPO and ALP positively correlated with bacterial and fungal communities associated with rhizosphere soils in this study [50]. We sequenced soil microbes on six altitude gradients in Mt. Shegyla on the Tibetan Plateau, and the results showed that the abundance of fungi decreased significantly with the increase in altitude, while the bacterial diversity showed a more significant decrease than fungi [51]. The PPO activity represents the degree of soil humification in the sample plot. Deforestation studies have shown that the activity of soil polyphenol oxidase decreases with increasing available nitrogen [52]. Pearson correlation analysis indicated that Proteobacteria and Bacteroidota showed significantly positive correlations with AP (*p* < 0.05), whereas AP was strongly negatively associated with Zygomycota (*p* < 0.05) (Appendix A and Appendix B ). Phosphorus is an important nutrient for plant growth and an important participant in plant life activities. In the alpine grasslands of northern Tibet, changes in pH, soil phosphorus and organic carbon play an important role in the composition of the arbuscular mycorrhizal (AM) fungal community [53]. In addition, AP also showed a significant positive relationship with Tibet fungi and bacterial communities (*p* < 0.05). Combined with the impact on the microbial community [54], we believe that AP was the key factor affecting the microbial community structure and diversity of *S. purpurea* in Tibet. Previous studies on the structure of plant microbial communities have shown that the composition and diversity of microbial communities are mainly affected by soil available phosphorus and total nitrogen [55], as well as organic matter and available water [56,57]. The results of our study further confirmed and expanded our understanding of the regulatory role of environmental factors on the distribution pattern of microbial communities.

## 5. Conclusions

A diverse array of fungi and bacteria inhabit the rhizosphere, root and leaf of *S. purpurea* in the QTP, and the composition and diversity of microbial communities in rhizosphere soil were higher than those in root and leaf samples. However, the composition and diversity of microbial communities in the same habitat from different sampling sites was insignificant. Habitat was the most important element affecting the microbial diversity of *S. purpurea*, compared to location. Soil environmental factors have modified the fungal and bacterial community structure of *S. purpurea*. Physicochemical properties and enzyme activity, such as DHA, SC, AK, AN, were the main driving factors affecting fungal microbial community. Soil physicochemical properties, especially TP and AN, were the main driving factors affecting the bacterial community structure of *S.purpurea*.

## Figures and Tables

**Figure 1 plants-11-00363-f001:**
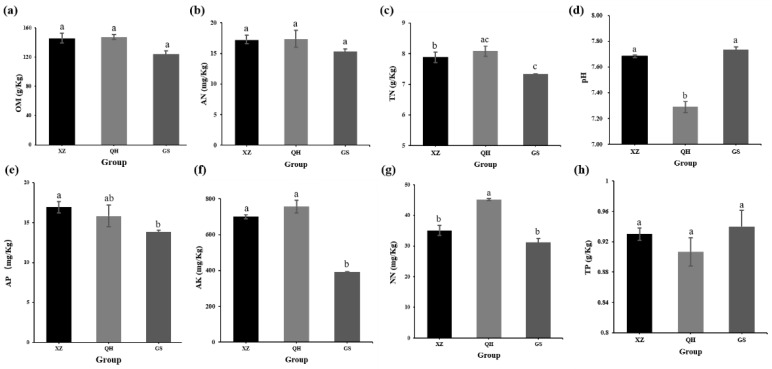
Changes in soil physicochemical properties. Organic matter (OM) (**a**), ammonium nitrate content (AN) (**b**), total nitrogen content (TN) (**c**), soil pH (**d**), available phosphate (AP) (**e**), available kalium (AK) (**f**), nitrate nitrogen content (NN) (**g**), total phosphorus content (TP) (**h**), in the sample of Tibet (Group XZ), Qinghai (Group QH), Gansu (Group GS). Different lowercase letters (a, b, c) indicate a significant difference at *p* ≤ 0.05.

**Figure 2 plants-11-00363-f002:**
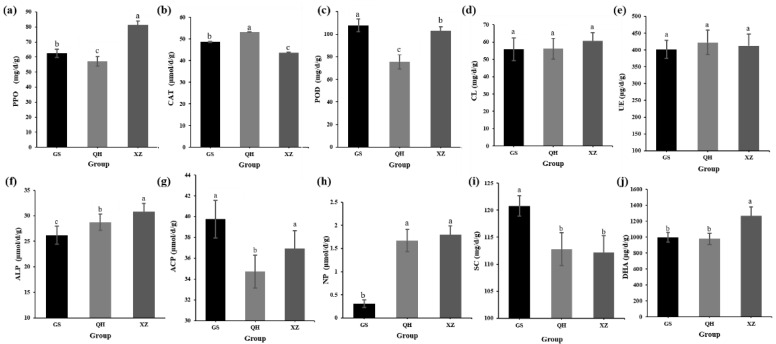
Changes in soil enzymatic activity. Polyphenol oxidase (PPO) (**a**), catalase (CAT) (**b**), peroxidase (POD) (**c**), cellulase (CL) (**d**), urease (UE) (**e**), alkaline phosphatase (ALP) (**f**), acid phosphatase (ACP) (**g**), neutral phosphatase (NP) (**h**), soil sucrase (SC) (**i**), dehydrogenase (DHA) (**j**). The samples were from Tibet (Group XZ), Qinghai (Group QH), Gansu (Group GS). Different lowercase letters (a, b, c) indicate a significant difference at *p* ≤ 0.05.

**Figure 3 plants-11-00363-f003:**
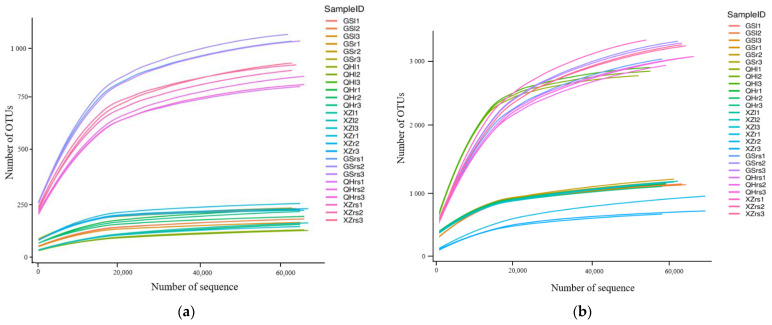
Rarefaction curve of fungal (**a**) and bacterial (**b**) communities.

**Figure 4 plants-11-00363-f004:**
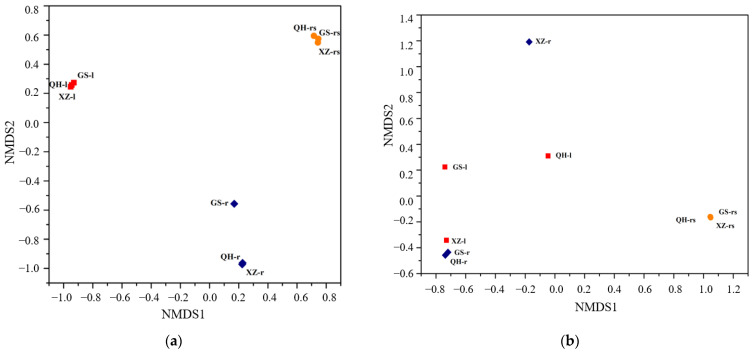
NMDS analysis of fungal (**a**) and bacterial (**b**) communities in rhizosphere soil, root and leaf of *S*. *purpurea* in different sampling sites.

**Figure 5 plants-11-00363-f005:**
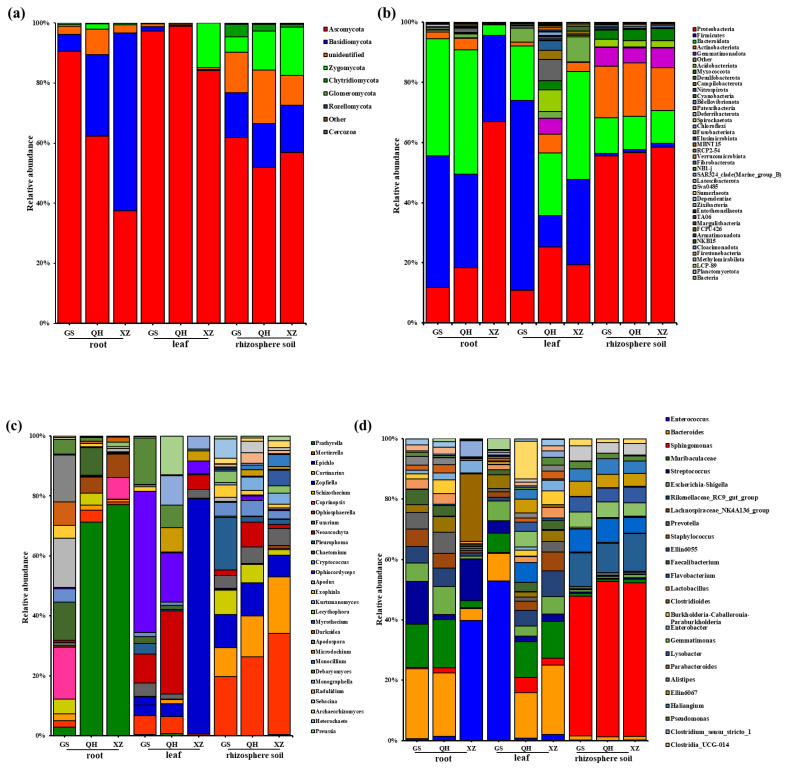
Relative abundance of fungal phyla (**a**), bacterial phyla (**b**), fungal genera (**c**), bacterial genera (**d**).

**Figure 6 plants-11-00363-f006:**
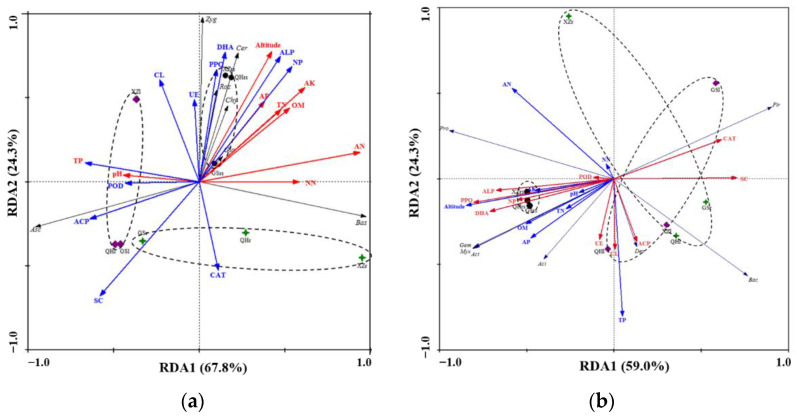
Correlations between soil properties and the community structure of fungi (**a**) and bacteria (**b**) as determined by redundancy analysis.

**Table 1 plants-11-00363-t001:** Diversity index of fungal and bacterial communities associated with *S*. *pur**purea*.

	Samples	Vaild Tags	OUT	Chao1	Shannon Index	Simpson Index	Coverage (%)
fungi	GSrs	64113	1037	1129.63 ± 9.75 ^aA^	7.21 ± 0.19 ^aA^	0.98 ± 0.01 ^aA^	99.76
	GSr	64583	227	254.50 ± 10.99 ^bB^	2.81 ± 0.01 ^bB^	0.55 ± 0.00 ^bB^	99.95
	GSl	66038	172	225.22 ± 6.90 ^cB^	1.06 ± 0.03 ^cC^	0.20 ± 0.01 ^cC^	99.94
	QHrs	66395	824	904.02 ± 27.80 ^aA^	6.78 ± 0.13 ^aA^	0.98 ± 0.00 ^aA^	99.80
	QHr	65696	208	246.75 ± 17.51 ^bB^	4.22 ± 0.01 ^bA^	0.89 ± 0.00 ^bB^	99.93
	QHl	66323	133	180.19 ± 27.34 ^cAB^	0.68 ± 0.05 ^cB^	0.13 ± 0.01 ^cC^	99.94
	XZrs	63755	907	1005.35 ± 16.59 ^aA^	6.93 ± 0.08 ^aA^	0.98 ± 0.00 ^aA^	99.77
	XZr	66549	232	256.44 ± 22.38 ^bB^	3.62 ± 0.04 ^bB^	0.78 ± 0.00 ^bB^	99.96
	XZl	66234	149	208.64 ± 20.68 ^cC^	2.18 ± 0.01 ^cC^	0.63 ± 0.00 ^cC^	99.93
bacteria	GSrs	60309	3131	3651.95 ± 64.65 ^aA^	9.50 ± 0.05 ^aA^	0.99 ± 0.00 ^aA^	98.62
	GSr	60100	1109	1411.07 ± 42.62 ^bB^	8.06 ± 0.00 ^bB^	0.99 ± 0.00 ^bB^	99.49
	GSl	62605	1056	1370.27 ± 15.11 ^cB^	5.49 ± 0.06 ^cC^	0.83 ± 0.01 ^cC^	99.56
	QHrs	62815	2926	3491.50 ± 14.90 ^aA^	9.33 ± 0.03 ^bB^	0.99 ± 0.00 ^aA^	98.68
	QHr	58611	1074	1398.16 ± 20.88 ^cB^	8.30 ± 0.02 ^cC^	0.99 ± 0.00 ^bB^	99.51
	QHl	53525	2858	3048.95 ± 49.73 ^bAB^	10.15 ± 0.02 ^aA^	1.00 ± 0.00 ^cC^	99.43
	XZrs	59571	3201	3731.81 ± 79.12 ^aA^	9.34 ± 0.06 ^bA^	0.99 ± 0.00 ^aA^	98.59
	XZr	64531	699	946.17 ± 174.39 ^cB^	2.58 ± 0.09 ^cC^	0.58 ± 0.01 ^bB^	99.58
	XZl	60692	1096	1439.46 ± 64.45 ^bB^	8.07 ± 0.02 ^bB^	0.99 ± 0.00 ^cC^	99.45

^a^ Significant differences represent comparisons between different sampling habitats at the same site; ^b^ data are means ± standard deviation (*n* = 3). Different lowercase letters (a, b, c) indicate a significant difference at *p* ≤ 0.05, whereas different uppercase letters (A, B, C) indicate a remarkable difference at *p* ≤ 0.01.

**Table 2 plants-11-00363-t002:** *p* value showing significance of the effects of plant growth elevations and habitats from Kruskal–Wallis one-way ANOVA on ranks (Tukey test) and Spearman rank order correlation for the fungal communities of *S. purpurea*.

Samples ^a^	Pairwise Comparison ^b^	Kruskal–Wallis One-Way ANOVA on Ranks		Tukey Test		Spearman Rank Order Correlation	
		**H**	** *p* **	**q**	** *p* **	**CC ^c^**	** *p* **
Rs	GS vs. QH	0.00197	0.965			0.624	*0.000232*
	GS vs. XZ	0.813	0.813			0.7	*≤0.0001*
	QH vs. XZ	0.0289	0.865			0.761	*≤0.0001*
L	GS vs. QH	0.139	0.709			0.467	*0.00961*
	GS vs. XZ	1.779	0.182			0.532	*0.00265*
	QH vs. XZ	3.221	0.073			0.732	*≤0.0001*
R	GS vs. QH	1.977	0.16			0.508	*0.00436*
	GS vs. XZ	3.221	0.073			0.462	*0.0105*
	QH vs. XZ	0.395	0.53			0.604	*≤0.0001*
GS	Rs vs. L	3.162	0.075			0.0809	0.669
	Rs vs. R	0.527	0.468			−0.137	0.466
	L vs. R	0.578	0.447			0.205	0.274
QH	Rs vs. L	1.385	0.239			0.116	0.539
	Rs vs. R	5.857	*0.016*	3.419	*<0.05*	0.0692	0.714
	L vs. R	1.039	0.308			0.463	*0.0103*
XZ	Rs vs. L	10.687	*0.001*	4.6	*<0.05*	−0.00771	0.966
	Rs vs. R	10.207	*0.001*	4.495	*<0.05*	−0.0712	0.705
	L vs. R	0.0819	0.775			0.168	0.373

Eighteen pairwise analyses were performed comprising each of the three possible pairings of the three different elevations and both collected samples (rhizosphere soil, leaves and roots) at the same elevation. Values in italics indicate significant differences or significant correlations. ^a^ Rs, rhizosphere soil; R, root; L, leaf. ^b^ GS, Gansu; QH, Qinghai; XZ, Tibet ^c^ CC, correlation coefficient.

**Table 3 plants-11-00363-t003:** Values showing significance of the effects of plant growth elevations and habitats from Kruskal–Wallis one-way ANOVA on ranks (Tukey test) and Spearman rank order correlation for the bacterial communities of *S. purpurea*.

Samples ^a^	Pairwise Comparison ^b^	Kruskal–Wallis One-Way ANOVA on Ranks		Tukey Test		Spearman Rank Order Correlation	
		**H**	** *p* **	**q**	** *p* **	**CC ^c^**	** *p* **
Rs	GS vs. QH	0.0315	0.859			0.972	≤0.0001
	GS vs. XZ	0.0219	0.882			0.947	≤0.0001
	QH vs. XZ	0.184	0.668			0.951	≤0.0001
L	GS vs. QH	6.694	*0.01*	3.659	*<0.05*	0.063	0.739
	GS vs. XZ	3.925	*0.048*	2.802	*<0.05*	0.746	≤0.0001
	QH vs. XZ	0.0789	0.779			0.187	0.319
R	GS vs. QH	0.0919	0.762			0.904	≤0.0001
	GS vs. XZ	1.181	0.277			0.575	*0.000958*
	QH vs. XZ	2.078	0.149			0.583	*0.000779*
GS	Rs vs. L	0.0177	0.894			−0.443	*0.0145*
	Rs vs. R	0.184	0.668			−0.27	0.147
	L vs. R	0.735	0.391			0.802	≤0.0001
QH	Rs vs. L	7.4	*0.007*	3.847	*<0.05*	0.495	*0.00568*
	Rs vs. R	1.399	0.237			−0.345	0.0619
	L vs. R	0.614	0.433			0.196	0.296
XZ	Rs vs. L	2.941	0.086			−0.355	0.0538
	Rs vs. R	0.184	0.668			−0.523	*0.00315*
	L vs. R	4.918	*0.027*	3.136	*<0.05*	0.695	≤0.0001

Eighteen pairwise analyses were performed comprising each of the three possible pairings of the three different elevations and both collected samples (rhizosphere soil, leaves and roots) at the same elevation. Values in italics indicate significant differences or significant correlations. ^a^ Rs, rhizosphere soil; R, root; L, leaf. ^b^ GS, Gansu; QH, Qinghai; XZ, Tibet. ^c^ CC, correlation coefficient.

## Data Availability

All relevant data are within this article.

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
