# Peer review of "Characterization of Rhizosphere and Endophytic Microbial Communities Associated with Stipa purpurea and Their Correlation with Soil Environmental Factors"

_plants, 2022, doi:10.3390/plants11030363_

Round 1

Reviewer 1 Report

I have some suggestes and coments to your manuscript, following:

•    The chapter "Material and methods" must be before the chapter "Results"
•    The chapter „Determination of soil physicochemical properties” must be appear before  the chapter 4.2” Determination of soil enzymatic activities”.
•    Line 94-95: The results showed that the pH of the soil at three sites was neutral, partially alkaline, but the lowest pH was found at the QH site compared to other experimental variants.
•    Line 94-98: This sentence is not in line with the statistical analysis and statistically significant differences present in the figure 1 :
„The contents of OM, AN, TN, AK and NN in QH 95 plot were higher than those in GS and XZ, and GS was the lowest (Figure.1 a, b, c, f, g). 96 But the TP showed an opposite increasing trend (Figure.1 h), which recorded no signifi- 97 cant difference. At the same time, a gradual increase was observed in AP as elevation 98 decreased (Figure.1e)”

The contents of OM, AN, AK and NN in QH  plot were higher than those in GS and XZ, and GS was the lowest (Figure.1 a, b, c, f, g). 96 But the TP showed an opposite increasing trend (Figure.1 h), which recorded no signifi- 97 cant difference. At the same time, a gradual increase was observed in AP as elevation 98 decreased (Figure.1e)”

The contens of OM, AN, TP is the same in the all samples (GS, XZ, QH) because there are the same letters.
•    Line 108-116: Figur 2 should be carefully analyzed and described, this text is not always corresponding with the statistical analysis and sometimes is not clear.

Author Response

Response to Reviewer 1 Comments

Point 1: The chapter "Material and methods" must be before the chapter "Results"

Response 1: Corrected accordingly.

Point 2: The chapter “ Determination of soil physicochemical properties” must be appear before the chapter 4.2” Determination of soil enzymatic activities”

Response 2: Corrected accordingly.

Point 3: Line 94-95: The results showed that the pH of the soil at three sites was neutral, partially alkaline, but the lowest pH was found at the QH site compared to other experimental variants.

Response 3: Thank you for your suggestion. A detailed modification was given accordingly, the relevant information showed up in lines 182-184.

Point 4: Line 94-98: This sentence is not in line with the statistical analysis and statistically significant differences present in the figure 1: The contents of OM, AN, TN, AK and NN in QH 95 plot were higher than those in GS and XZ, and GS was the lowest (Figure.1 a, b, c, f, g). 96 But the TP showed an opposite increasing trend (Figure.1 h), which recorded no signifi- 97 cant difference. At the same time, a gradual increase was observed in AP as elevation 98 decreased (Figure.1e)”. The contents of OM, AN, TP is the same in the all samples (GS, XZ, QH) because there are the same letters.

Response 4: Thank you for your criticism. A detailed explanation was given accordingly, the relevant information showed up in lines 184-188.

Point 5: Line 108-116: Figure 2 should be carefully analyzed and described, this text is not always corresponding with the statistical analysis and sometimes is not clear.

Response 5: Thank you for your criticism. A detailed explanation was given accordingly, the relevant information showed up in lines 196-205.

Reviewer 2 Report

Presented manuscript is describes new data on specifics of microbial communities associated with Stipa purpurea in Qinghai-Tibetan Plateau. The manuscript performed well: all the sections looks necessary and logically complete.  I have just a several editorial remarks to make the manuscript bit more reader-friendly.

line 20 were dominant in microbial...

lines 23-24 - Selected sentence - some explanation of the observed OTU abundance in the different types of plant should be mentioned.

line 71 - space is missed

All figures should be presented as vector graphic elements (for example, as PDF-files) to provide good resolution and readers' comfort.

line 140 Shannon should starts with capital

lines 221 - 229 - Selected genera names should be italics

Author Response

Response to Reviewer 2 Comments

Point 1: line 20 were dominant in microbial...

Response 1: We have added “ in ”.

Point 2: lines 23-24 - Selected sentence - some explanation of the observed OTU abundance in the different types of plant should be mentioned.

Response 2: Thank you for your suggestion. A detailed explanation was given accordingly, accordingly, the relevant information showed up in lines 25-27 and lines 216-224.

Point 3: line 71 - space is missed

Response 3: We have modified the format.

Point 4: All figures should be presented as vector graphic elements (for example, as PDF-files) to provide good resolution and readers' comfort.

Response 4: Thank you for your suggestion. Corrected accordingly.

Point 5: line 140 Shannon should starts with capital

Response 5: Corrected accordingly.

Point 6: lines 221 - 229 - Selected genera names should be italics

Response 6: Corrected accordingly.

Reviewer 3 Report

Dear authors,

I have read with interest the manuscript "Characterization of rhizosphere and endophytic microbial communities associated with Stipa purpurea and their correlation with soil environmental factors". The topic and the idea of your research is interesting, and I propose you some changes in order to improve it.

Abstract - remove the word "we" and make it more impersonal. In this way, you can provide a more generic form. Make the sentence shorter - 2-3 rows only. This will point better each of your findings. This suggestion you should take into account in the entire text.

Results section - expand the interpretation of your results. For example, sub-section 2.1 have only 6 rows of results and the figure 5 rows. Please consider expanding the results. For figure 1 - maintain only p<0.05. It is hard to read a figure with 2 p.values. The same suggestions for subsection 2.2, present the results in each figure separately (a, b, c) in a sentence. This will be helpful when you write the discussion.

For figure 6 add the variance explained by each axis, if available.

Discussion section - remove references to previous presented results. These sentences should be moved to Results section. In this section, you should compare your findings with other international works. Add more international references in the same area, this will sustain the unique character of your work.

Conclusion - this section should be rewritten. Here you present your main findings, with values where is the case.

Author Response

Response to Reviewer 3 Comments

Point 1: Abstract - remove the word "we" and make it more impersonal. In this way, you can provide a more generic form. Make the sentence shorter - 2-3 rows only. This will point better each of your findings. This suggestion you should take into account in the entire text.

Response 1: Thank you for your criticism. A detailed explanation was given accordingly, the relevant information showed up in lines 19-21.

Point 2: Results section - expand the interpretation of your results. For example, sub-section 2.1 have only 6 rows of results and the figure 5 rows. Please consider expanding the results. For figure 1 - maintain only p<0.05. It is hard to read a figure with 2 p.values. The same suggestions for subsection 2.2, present the results in each figure separately (a, b, c) in a sentence. This will be helpful when you write the discussion.

Response 2: We agree with the criticism. We have extended the results section and maintained only p<0.05 in the figure 1 and 2. The relevant information showed up in manuscript lines 181-205.

Point 3: For figure 6 add the variance explained by each axis, if available.

Response 3: we have added the variance explained by each axis.

Point 4: Discussion section - remove references to previous presented results. These sentences should be moved to Results section. In this section, you should compare your findings with other international works. Add more international references in the same area, this will sustain the unique character of your work.

Response 4: Corrected accordingly.

Point 5: Conclusion - this section should be rewritten. Here you present your main findings, with values where is the case.

Response 5: Thank you for your criticism. A detailed explanation was given accordingly, the relevant information showed up in lines 462-471.

Round 2

Reviewer 1 Report

Everything is corrected according to my comments.

Reviewer

Reviewer 3 Report

Dear authors, you have improved the form of the article. This one is more interesting to readers.